# Electronic Nose Drift Suppression Based on Smooth Conditional Domain Adversarial Networks

**DOI:** 10.3390/s24041319

**Published:** 2024-02-18

**Authors:** Huichao Zhu, Yu Wu, Ge Yang, Ruijie Song, Jun Yu, Jianwei Zhang

**Affiliations:** School of Control Science and Engineering, Dalian University of Technology, Dalian 116000, China; zhuhuichao@dlut.edu.cn (H.Z.); wyrose@mail.dlut.edu.cn (Y.W.); g32109306@mail.dlut.edu.cn (G.Y.); songruijie@mail.dlut.edu.cn (R.S.); junyu@dlut.edu.cn (J.Y.)

**Keywords:** anti-drift, electronic nose, transfer learning, deep learning

## Abstract

Anti-drift is a new and serious challenge in the field related to gas sensors. Gas sensor drift causes the probability distribution of the measured data to be inconsistent with the probability distribution of the calibrated data, which leads to the failure of the original classification algorithm. In order to make the probability distributions of the drifted data and the regular data consistent, we introduce the Conditional Adversarial Domain Adaptation Network (CDAN)+ Sharpness Aware Minimization (SAM) optimizer—a state-of-the-art deep transfer learning method.The core approach involves the construction of feature extractors and domain discriminators designed to extract shared features from both drift and clean data. These extracted features are subsequently input into a classifier, thereby amplifying the overall model’s generalization capabilities. The method boasts three key advantages: (1) Implementation of semi-supervised learning, thereby negating the necessity for labels on drift data. (2) Unlike conventional deep transfer learning methods such as the Domain-adversarial Neural Network (DANN) and Wasserstein Domain-adversarial Neural Network (WDANN), it accommodates inter-class correlations. (3) It exhibits enhanced ease of training and convergence compared to traditional deep transfer learning networks. Through rigorous experimentation on two publicly available datasets, we substantiate the efficiency and effectiveness of our proposed anti-drift methodology when juxtaposed with state-of-the-art techniques.

## 1. Introduction

Gas sensors play a vital role across diverse industrial sectors, including environmental surveillance [1,2,3], medical diagnostics [4,5,6], food analytics [7,8], and explosive detection [9,10]. Over the past two decades, significant strides have been made in gas sensor technology to meet the practical demands of various applications. For instance, Fort and colleagues proposed three measurement methodologies to effectively differentiate gas mixtures [11], enabling a more precise categorization of wines. This empowers industries to ensure the quality and authenticity of their products. Bhattacharyya et al. introduced a computational framework integrating a cost-effective interface and a wide-range, low-value resistive sensor [12,13]. This architecture can assess the quality of unidentified tea samples, providing an economical and efficient solution for the tea industry. In another notable development, Brezmes et al. designed a sensor system specifically for measuring fruit ripeness, tailored to application-specific requirements [14]. This system enables a precise and timely evaluation of fruit maturity, assisting in the optimization of harvesting and storage operations. In summary, advancements in gas sensor technology have significantly improved the capability to detect and analyze gases across various industries. These innovations have led to more accurate and reliable outcomes, ultimately enhancing productivity and safety in these sectors. However, since the measurement strategy of gas sensors is to detect the change in resistance and voltage of the gas-sensitive material when it is exposed to the gas to be measured, the sensor sensitivity can be affected by various aspects such as temperature, humidity, pressure, self-aging, and poisoning. Changes in sensor sensitivity can lead to fluctuations in sensor response when the electronic nose is exposed to the same gas at different times, called sensor drift [15]. This paper focuses on the drift compensation of gas sensors.

In order to tackle this dilemma, researchers have approached it from three different perspectives. The first approach involves developing gas-sensitive materials that exhibit both high performance and high stability. However, this necessitates breakthroughs in multiple disciplines like physics, chemistry, and materials science, and can be quite costly. Another approach involves enhancing the stability of the gas sensor by modifying its operating mode, such as periodically adjusting the heating voltage. Nevertheless, these two strategies mainly address short-term drift phenomena and have limited impact on long-term drift issues.

To combat long-term drift problems, many researchers have focused on modifying the signal-processing algorithms used in gas sensors. These algorithms are typically classified into three groups: data-level, feature-level, and classifier-level drift compensation methods.

Data-level approaches: Artursson et al. introduced techniques such as Principal Component Analysis (PCA) and Partial Least Squares for drift suppression [16]. Padilla et al. presented an OSC-based drift correction strategy for gas sensor arrays [17]. Natale et al. addressed drift by employing Independent Component Analysis (ICA) while preserving components associated with sample characteristics [18]. Additionally, a method known as Common Principal Component Analysis (CCPCA) offers drift reduction without requiring a distinct reference gas [19].Feature-level methods: These approaches aim to align source data (clean data) and target data (drift data) in a shared subspace, minimizing distribution divergence between them. L. Zhang proposed Domain Regularized Component Analysis (DRCA), which reduces marginal distribution divergence between clean and drift data within the common subspace [20]. An extension of DRCA, Local Discriminant Subspace Projection (LDSP), seeks to identify a common subspace that simultaneously reduces local within-class variance of projected source samples and maximizes local between-class variance [21]. Another approach, named Common Subspace-Based Drift (CSBD), minimizes distribution divergence between clean and drift data within a new subspace [22].Classifier-level techniques: The performance of a classifier significantly impacts the resulting classification [23]. Zhang and Zhang introduced two gas drift correction methods based on Extreme Learning Machines, both of which provide low computational complexity [24]. In recent years, online drift compensation methods have been introduced to address sensor drift [25,26,27]. Expanding on the concept of active learning, the method (referred to as AL-ISSMK) developed by Liu et al. [26] identifies the most valuable samples and retrains the classifier to adapt to evolving sensor drift.

While the adaptive correction methods mentioned above have shown promising results in compensating for drift in gas sensor arrays, there remain three areas that require further enhancement: (1) Low classification accuracy persists, with most methods achieving rates below 90%. (2) Many approaches rely on labeled data from drifted sensors to enhance accuracy, but obtaining these labels is costly as it involves recalibrating the sensors. (3) Several methods necessitate an excessive number of hyperparameters, limiting their practicality for real-world applications in production and daily life.

To address the previously mentioned challenges, we present the CDAN+SAM model. In this model, CDAN is devised to extract common features from both clean and drifted data. These extracted features are subsequently input into a neural network to train a more generalized and robust classifier. The SAM optimizer plays a crucial role in smoothing the training process, facilitating easier network training and convergence. The fundamental structure of the CDAN+SAM model is illustrated in Figure 1.

The remainder of this paper is organized as follows: The second section provides an introduction to the foundational theory of transfer learning, offering insights into the principles underlying CDAN and SAM. In the third section, we conduct a comprehensive analysis of experimental results and perform ablation experiments to further validate our approach. Finally, the fourth section summarizes the key findings and conclusions of this paper.

## 2. Theoretical Background

### 2.1. Transfer Learning

The domain and task represent the foundational concepts in transfer learning. In this context, given a source domain ($D_{S}$) paired with a corresponding source task ($T_{S}$) and a target domain ($D_{T}$) with its associated task ($T_{T}$), transfer learning aims to enhance the predictive function $f_{T}\left(\right)$ for the target by leveraging relevant information from $D_{S}$ and $T_{S}$, where $D_{S}\neq D_{T}$ or $T_{S}\neq T_{T}$ [28].

Evidently, the target domain $D_{T}$ (drift data) and the source domain $D_{S}$ (clean data) exhibit differences in their feature distributions due to sensor drift. Consequently, a classifier trained on clean data becomes unreliable when applied to drift data. Despite both domains measuring the same gas, and thus sharing the same category space ($Y_{s}=Y_{t}$), inconsistencies arise in the marginal and conditional probability distributions between the two domains. The objective of transfer learning is to train a classifier using clean data to accurately predict the labels of drift data.

### 2.2. Conditional Adversarial Domain Adaptation Network (CADN)

Deep transfer learning has emerged as a prominent research direction within the field of transfer learning. Researchers are increasingly focused on training domain-invariant classifiers in deep networks to enhance the generalization capabilities of transfer learning methods across diverse data distributions. Adversarial learning has been integrated into deep networks to facilitate the learning of disentangled and transferable representations for domain adaptation. In comparison to other deep transfer methods, conditional adversarial domain adaptation considers not only the inherent correlation within the original data but also the relationships between different categories.

This method is conceptualized as a minimax optimization problem involving two competing error terms: (a) Minimizing the error for classifiers generated from source domain data and source domain labels ensures improved classifier performance on the source domain data. (b) Maximizing the error generated by a domain discriminator trained with both source and target data is designed to confuse the discriminator regarding whether the data originates from the source or target domain.
(1)$$\begin{array}{c} E_{C}=\frac{1}{n_{s}}\sum_{i=1}^{n_{s}}L\left(C\left(f_{i}^{s}\right),y_{i}^{s}\right)\\ E_{D,C}=-\frac{1}{n_{s}}\sum_{i=1}^{n_{s}}\log\left[D\left(f_{i}^{s},c_{i}^{s}\right)\right]-\frac{1}{n_{t}}\sum_{j=1}^{n_{t}}\log\left[1-D\left(f_{j}^{t},c_{j}^{t}\right)\right] \end{array}$$

The optimization objective poses an extreme value optimization problem for training the feature extraction model G, aiming to minimize empirical risk on the source domain data and reduce classification errors on the same data. Simultaneously, the trained feature extraction model G is required to maximize the loss incurred by the domain discriminator model. In the training of the discriminator D, it is crucial for D to create confusion, making it challenging to determine whether the samples are from the source domain dataset or the target domain dataset. The entropy of the domain discrimination model serves as a quantitative measure of the sample migration performance.
(2)$$\begin{array}{cc} \; & \min_{C}E_{C}-\lambda E_{D,C}\\ \; & \min_{D}E_{D,C} \end{array}$$

Additionally, conditional entropy is employed as a metric for migrability, and the entropy of the sample prediction vector is utilized as the migration weight for the input of the domain discriminant model. Conditional adversarial domain adaptation asserts that the migration performance of a sample is reflected in its category confidence, with samples exhibiting higher category confidence (more clearly labeled) demonstrating superior migration performance. The entropy of the domain discrimination result is also incorporated as a weight for the classification loss originating from the source domain samples.
(3)$$\begin{array}{c} E_{C}=\frac{1}{n_{s}}\sum_{i=1}^{n_{s}}L\left(C\left(f_{i}^{s}\right),y_{i}^{s}\right)\\ E_{D,C}=-\frac{1}{n_{s}}\sum_{i=1}^{n_{s}}e^{-H(c_{j}^{s})}\log\left[Dh_{i}^{s}\right)]-\frac{1}{n_{t}}\sum_{j=1}^{n_{t}}e^{-H(c_{j}^{t})}\log\left[1-D\left(h_{j}^{t}\right)\right] \end{array}$$

At this juncture, we have formulated the objective function for transfer weight-based conditional adversarial domain adaptation, which shares a similar structure with the generative adversarial model. Notably, there are two distinctive features: (1) The predicted category vectors are initially applied to enhance the performance of the domain discriminative model. (2) The predicted category vector serves as a metric for sample mobility at the input of the domain discrimination model.
(4)$$\begin{array}{cc} \; & \min_{C}\frac{1}{n_{s}}\sum_{i=1}^{n_{s}}L\left(C\left(x_{i}^{s}\right),y_{i}^{s}\right)-\frac{\lambda }{n_{s}}\sum_{i=1}^{n_{s}}e^{-H\left(c_{i}^{s}\right)}\log\left[D\left(h_{i}^{s}\right)\right]-\frac{\lambda }{n_{t}}\sum_{j=1}^{n_{t}}e^{-H\left(c_{j}^{t}\right)}\log\left[1-D\left(h_{j}^{t}\right)\right]\\ \; & \max_{D}\frac{1}{n_{s}}\sum_{i=1}^{n_{s}}e^{-H\left(c_{i}^{s}\right)}\log\left[D\left(h_{i}^{s}\right)\right]+\frac{1}{n_{t}}\sum_{j=1}^{n_{t}}e^{-H\left(c_{j}^{t}\right)}\log\left[1-D\left(h_{j}^{t}\right)\right] \end{array}$$

Among various factors, λ represents the trade-off hyperparameter balancing the source domain classification loss and domain discrimination loss. The joint variable h=(c,f) integrates the feature vector *f* and the category prediction vector *c* for a specific domain, commonly achieved through a multilinear operation denoted as h=f⊗c. The structural disparity between the conditional adversarial domain adaptation network and the traditional domain adversarial network is illustrated in Figure 2. In the traditional domain adversarial network, the feature is directly fed into the domain discriminator, whereas the conditional adversarial network inputs a cross product of the prediction vector and the feature vector into the domain discriminator. The entropy of the prediction vector (depicted by the dashed line) is also utilized as a weight for adversarial loss, emphasizing the portions more likely to undergo migration.

### 2.3. Smoothness in Domain Adversarial Training

Recently, numerous studies have explored the implications of integrating formulations that enhance smoothness into the domain adversarial training framework. This methodology incorporates a dual objective, comprising the primary task’s loss (such as classification or regression) and adversarial components. Researchers have observed that striving for convergence towards a smooth minimum with respect to the task loss stabilizes the adversarial training process, leading to enhanced performance in the target domain. Conversely, their analysis suggests that pursuing convergence towards smooth minima in adversarial loss may result in suboptimal generalization in the target domain.

Building on these insights, we introduce the Sharpness Aware Minimization (SAM) optimizer, a methodology designed to effectively boost the performance of domain adversarial methods in the context of electronic nose system compensation tasks. The fundamental idea behind SAM is to identify a smoother minimum (i.e., low loss in the ϵ neighborhood of θ) by utilizing the following formally defined objective:(5)$$\min_{\theta }\max_{∥\epsilon ∥\leq \rho }L_{obj}\left(\theta +\epsilon \right)$$

Here, $L_{obj}$ represents the objective function to be minimized, and ρ≥0 is a hyperparameter that sets the maximum norm for ϵ. Given the inherent difficulty in obtaining the exact solution for the inner maximization, SAM maximizes the first-order approximation instead:(6)$$\begin{array}{c} \hat{\epsilon }\left(\theta \right)\approx \underset{∥\epsilon ∥\leq \rho }{\arg\max}L_{obj}\left(\theta \right)+\epsilon^{T}\nabla_{\theta }L_{obj}\left(\theta \right)\\ =\rho \nabla_{\theta }L_{obj}\left(\theta \right)/{∥\nabla_{\theta }L_{obj}\left(\theta \right)∥}_{2} \end{array}$$

The term $\hat{\epsilon }\left(\theta \right)$ is incorporated into the weights θ. The gradient update for θ is subsequently computed as ${\left.\nabla_{\theta }L_{obj}\left(\theta \right)\right|}_{\theta +\hat{\epsilon }\left(\theta \right)}$. The outlined procedure can be regarded as a universal smoothness-enhancing formulation applicable to any $L_{obj}$. Now, we similarly introduce the concept of sharpness-aware source risk to identify a smooth minimum:(7)$$\max_{∥\epsilon ∥\leq \rho }R_{S}^{l}\left(h_{\theta +\epsilon }\right)=\max_{∥\epsilon ∥\leq \rho }E_{x∼P_{S}}\left[l\left(h_{\theta +\epsilon }\left(x\right),f\left(x\right)\right)\right]$$

We articulate the optimization objective of the proposed Smooth Domain Adversarial Training as follows:(8)$$\min_{\theta }\max_{\Phi }\max_{∥\epsilon ∥\leq \rho }E_{x∼P_{S}}\left[l\left(h_{\theta +\epsilon }\left(x\right),y\left(x\right)\right)\right]+d_{S,T}^{\Phi }$$

The first term represents the sharpness-aware risk, while the second term corresponds to the discrepancy term, which, notably, lacks smoothness in our approach.The flowchart of the CDAN+SAM implementation is shown in Figure 3.

## 3. Result and Discussion

To assess the efficacy of CDAN+SAM, we conducted a comparative analysis with various deep transfer learning methods using two publicly available sensor drift datasets as benchmarks. Resnet served as the feature extraction network in this model. The experimental configurations are delineated in the subsequent subsections. The computational environment utilized Pycharm, and the hardware specifications are as follows: Windows 10 operating system, Intel Core i7-10300H CPU @ 3.40 GHz, 32.0 GB RAM, GTX 3080 GPU, and a 2 TB SSD.

### 3.1. Experiment on Sensor Drift Dataset A

Dataset A used in Experiment 1 is from UCSD [23], and the dataset measures 6 types of gases, using 16 gas sensors (TGS2600, TGS2602, TGS2610, and TGS2620; 4 of each sensor). The dataset has 8 dimensional features per sample, including 2 rising edge features, 3 falling edge features, and 3 smooth states, and contains a total of 13,910 samples divided into 10 batches. The data were recorded from January 2008 to the end of February 2011, spanning 3 years, where Table 1 shows the details of the dataset and the scatter plot in Figure 4 shows the principal component analysis(PCA) of the dataset. We take Batch 1 as the source domain for model training and test on Batch K, K = 2, …, 10 (target domains). The classification accuracy on Batch K is reported.

In order to verify the effectiveness of the algorithms, 14 methods of 3 types, namely, drift compensation methods, traditional transfer learning methods, and deep transfer learning methods, are selected for comparison in this paper, of which SVM-rbf, OSC, CC-PCA, GLSW [29], DS [30], and DRCA belong to the drift compensation methods, and these types of methods are capable of identifying and calibrating drift components, and geodesic flow kernel (GFK) [31], TCA [32] and JDA [33] belong to the traditional migration learning methods, which can change the probability distribution of the data in order to improve the recognition algorithm accuracy. Deep Transfer Learning Methods: Within this category are DANN [34], WDANN [35], and MADA [36]. These methods represent mainstream approaches for deep domain adaptation. Experiments were conducted on sensor drift Dataset A, and the recognition results for different methods under the experimental setting are presented in Table 2 and Figure 5. It is observed that the proposed CDAN+SAM achieves the best classification performance. The average classification accuracy is 90.32%, which is 7.27% higher than the second-best learning method.

Furthermore, for each batch, the best parameters for which the proposed method achieves the highest accuracy are provided in Table 3. The feature extraction network is the Resnet18 network. Since the features of Dataset A are 128 dimensional, a deeper network is needed to extract the features.

### 3.2. Experiment on Sensor Drift Dataset B

The drift displacement electronic nose dataset was collected by Zhang Lei et al. from Chongqing University [20]. The dataset was collected using an array of electronic nose sensors of the same model. Experimental measurements included ammonia, benzene, carbon monoxide, formaldehyde, nitrogen dioxide, and toluene. And four TGS series (TGS2602, TGS2620, TGS2201A, and TGS2201B) air sensors were used as well as temperature and humidity sensors (STD2230-I2 Cof Sensirion in Switzerland). The dataset has 6-dimensional features for each sample, and contains a total of 1604 samples, divided into 3 batches: master data, Slave data 1, and Slave data 2, where the master data was collected 5 years prior to Slave 1 data and Slave 2 data. Table 4 records the detailed data of this dataset. The scatter plot in Figure 6 shows the principal component analysis(PCA) of the dataset. Notably, the distributions of the slave systems differ significantly from those of the master system.

We used the master data as the source domain of the model and the Slave 1 and Slave 2 datasets as the target domain of the model. The proposed CDAN+SAM is compared with 11 popular transfer learning methods, and the classification results are presented in Table 5 and Figure 7. It is evident that CDAN+SAM consistently demonstrates optimal identification accuracy. Specifically, when compared with WDAAN, which exhibits similarity to the proposed method, CDAN+SAM improves the average recognition rates by 6.21% and 13.82% for Tasks 1 and 2, respectively.

Furthermore, for each batch, the best parameters leading to the highest accuracy for the proposed method are detailed in Table 6. The feature extraction network is a CNN network. Since the features of this dataset are 6-dimensional, no deeper network is needed to extract the features.

### 3.3. The Sensitivity of CDAN+SAM to Different Magnitudes of Drift

CDAN+SAM achieves more than 85% accuracy for the first 7 batches of data in Dataset A and for 3 years, which indicates that the method can compensate the accuracy of short-term drift well. For the last 3 batches of data and for more than 2 years, except for Dataset 9, the accuracy of the compensation is mostly lower than 80% due to the serious drift of the dataset, but it is still higher than that of the other 12 methods. This indicates that CDAN+SAM can handle both short-term and longer-term drifts well.

Compared with the Dataset A, Dataset B has a larger time span and deeper drift, so the average compensation accuracies obtained by all the methods in Dataset B are lower than those obtained by the methods in Dataset A. However, CDAN+SAM achieves the best results in both slaves, which shows that the method can deal with more complex and deeper drift scenarios.

### 3.4. Ablation Study

To comprehensively analyze the role of the SAM component in CDAN+SAM, we conducted ablation experiments under two settings on both Dataset A and Dataset B utilizing CDAN+SAM.

Setting 1: To demonstrate the importance of CDAN in extracting features common to both source and target data, the term CDAN in CDAN+SAM was replaced with DANN. DANN, in contrast to CDAN, solely considers the distinctions between source and target domain data, overlooking the differences between various categories within the data.

Setting 2: To illustrate that the SAM optimizer contributes to smoothing the entire model for improved results, the SAM optimizer in CDAN was replaced with the SGD optimizer.

The results of the ablation experiments for these two settings are summarized in Table 7 and Table 8. Ablation study histograms of accuracy under Dataset A and Dataset B are visualized in Figure 8 and Figure 9.

The ablation study outcomes highlight that each component plays a crucial role in enhancing the domain adaptation capability of the CDAN+SAM model. The experiments emphasize that, in deep transfer learning, consideration should be given not only to the distinctions between the source and target domain data but also to the differences among various categories within the data. Furthermore, the SAM optimizer proves effective in smoothing the adversarial model, leading to superior results.

## 4. Conclusions

This paper presents a novel framework CDAN+SAM for gas sensor drift compensation. Traditional machine learning approaches face challenges in solving the sensor drift problem, which is mainly attributed to the aging of gas-sensitive materials leading to inconsistencies in the probability distributions of calibration and measurement data. In this case, the proposed CDAN+SAM framework excels in capturing the common features of the drifted and raw data, as the model considers not only the relationship between the drifted and clean data, but also the relationship between the data of different species of gases. The SAM optimizer used in CDAN+SAM mitigates the challenges associated with the traditional deep migration learning, such as the training difficulty and the convergence problems. Experimental results demonstrate the superior performance of CDAN+SAM, which outperforms most of the existing methods in long-term and short-term drift scenarios by improving the accuracy by 7.27% and 10.02%, respectively. We plan to use the CDAN+SAM method in real life in the future, which should use different feature extraction networks when dealing with different drifting datasets, e.g., for datasets with temporal features, the LSTM network can be used; for complex and huge datasets, the Transformer network can be used. The use of different networks will inevitably lead to a huge overhead of computational resources, so we suggest that the sensors should be deployed with 5G network data transmission devices, and cloud computing can be used to solve the problem of insufficient computational resources.

## Figures and Tables

**Figure 1 sensors-24-01319-f001:**
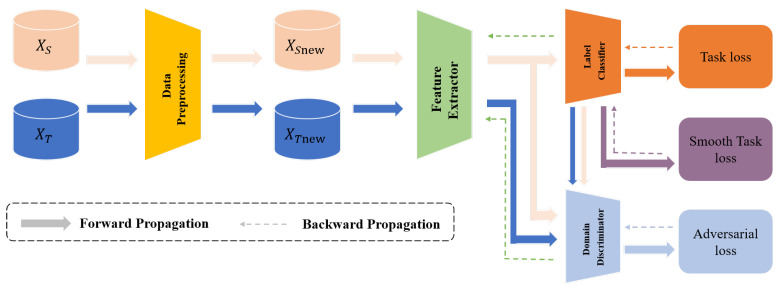
The basic structure of the CDAN+SAM model.

**Figure 2 sensors-24-01319-f002:**
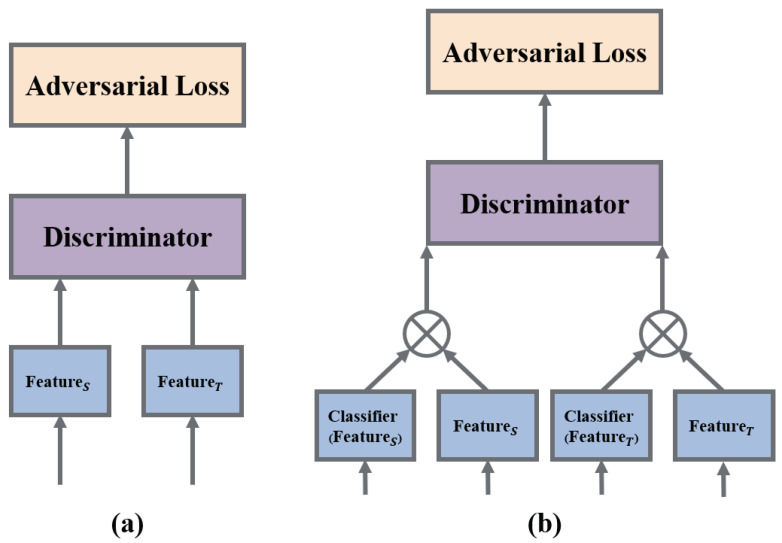
(**a**) The structure of the traditional domain adversarial loss. (**b**) The structure of the conditional adversarial loss.

**Figure 3 sensors-24-01319-f003:**
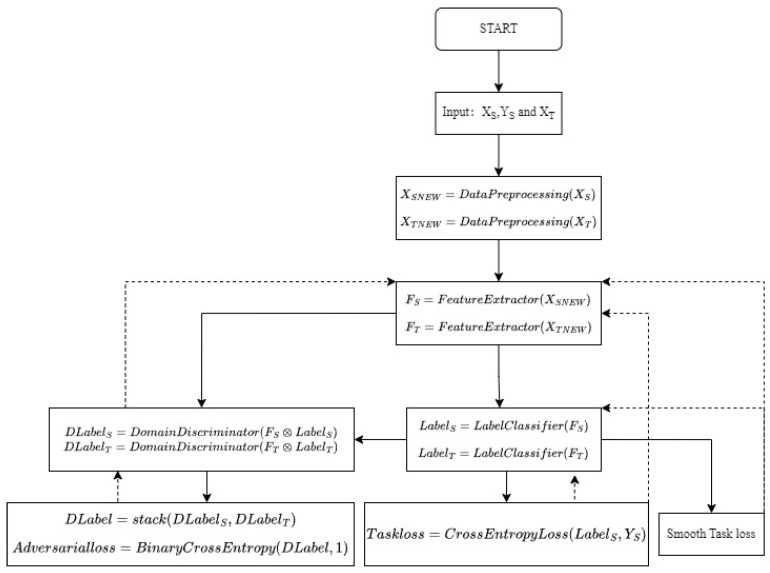
The flowchart of the CDAN+SAM.

**Figure 4 sensors-24-01319-f004:**
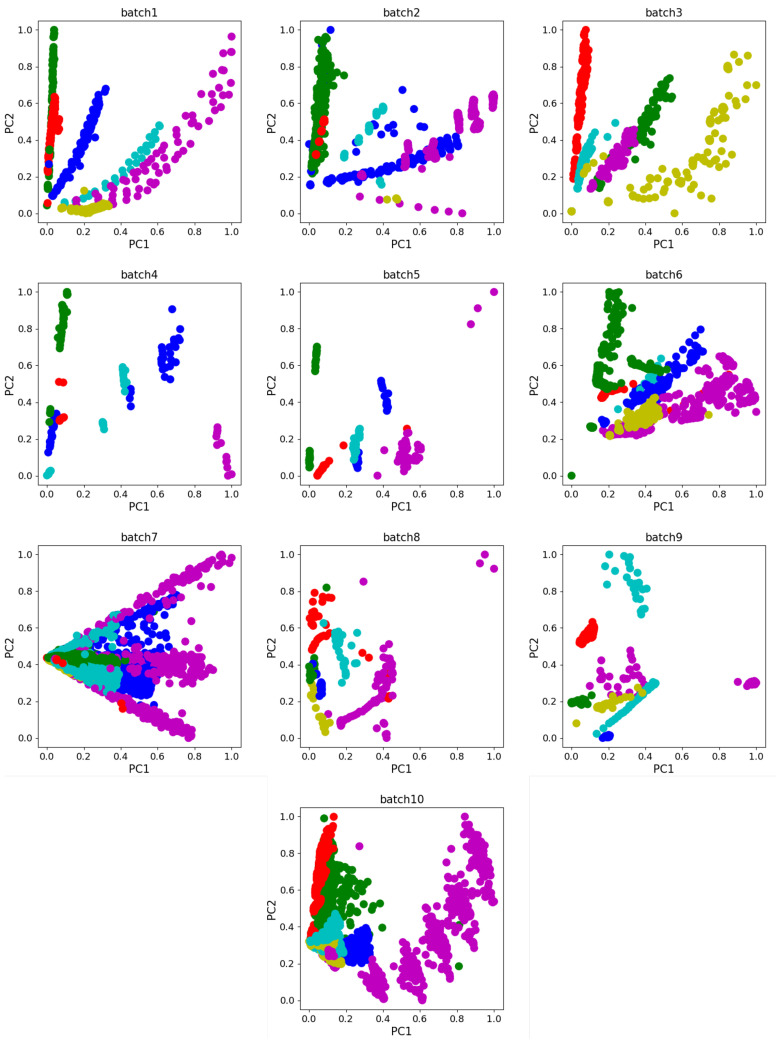
PCA scatter diagram of Dataset A.

**Figure 5 sensors-24-01319-f005:**
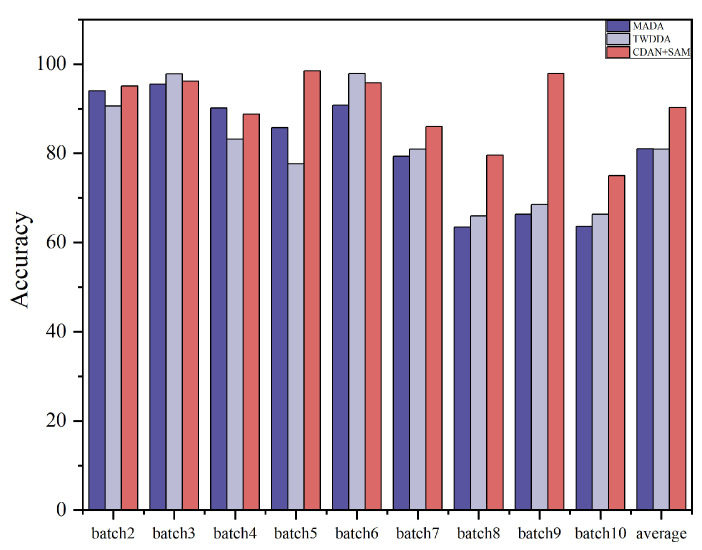
Histogram of the recognition effects of some of the algorithms in Dataset A.

**Figure 6 sensors-24-01319-f006:**
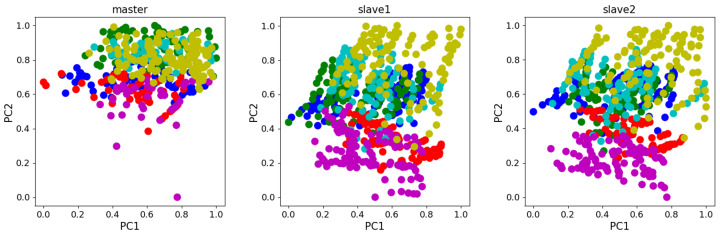
PCA scatter diagram of Dataset B.

**Figure 7 sensors-24-01319-f007:**
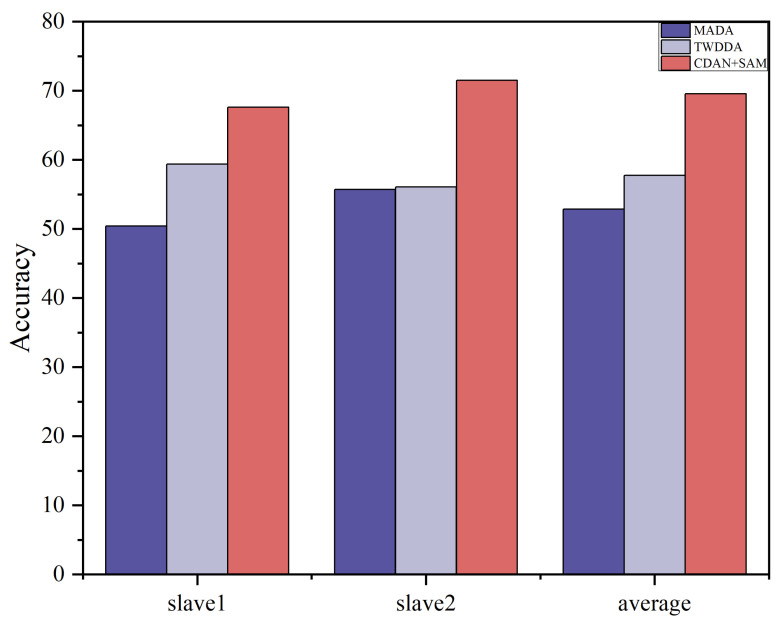
Histogram of the recognition effects of some of the algorithms in Dataset B.

**Figure 8 sensors-24-01319-f008:**
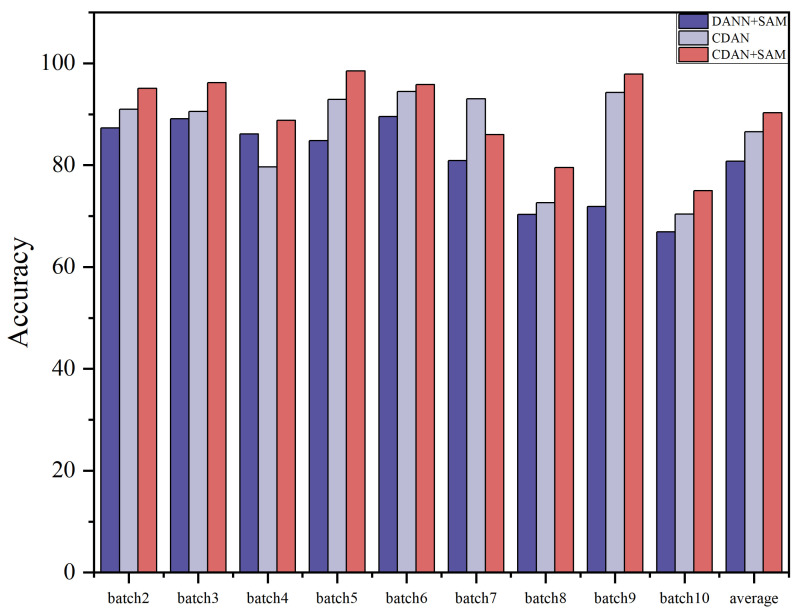
Histogram of accuracy in ablation study Dataset A.

**Figure 9 sensors-24-01319-f009:**
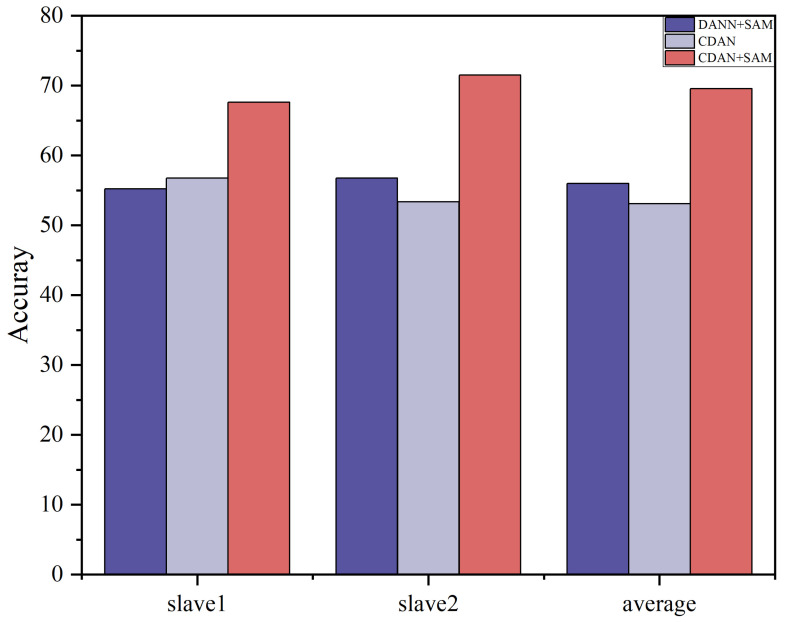
Histogram of accuracy in ablation study Dataset B.

**Table 1 sensors-24-01319-t001:** Benchmark sensor drift dataset from UCSD.

Batch	Month	C_2_H_5_OH	C_2_H_4_	NH_3_	CH_3_CHO	C_3_H_6_O	C_7_H_8_	Total
1	1, 2	83	30	70	98	90	74	445
2	3–10	100	109	532	334	164	5	1244
3	11–13	216	240	275	490	365	0	1586
4	14, 15	12	30	12	43	64	0	161
5	16	20	46	63	40	28	0	197
6	17–20	110	29	606	574	514	467	2300
7	21	260	744	630	662	649	568	3613
8	22, 23	40	33	143	30	30	18	294
9	24, 30	100	75	78	55	61	101	470
10	36	600	600	600	600	600	600	3600

**Table 2 sensors-24-01319-t002:** Recognition accuracy (%) under Dataset A.

Method	1–2	1–3	1–4	1–5	1–6	1–7	1–8	1–9	1–10	Average
PCASVM	82.40	84.80	80.12	75.13	73.57	56.16	48.64	67.45	49.14	68.60
SVM-rbf	74.36	61.03	50.93	18.27	28.26	28.81	20.07	34.26	34.47	38.94
SVM-gfk	72.75	70.08	60.75	75.08	73.82	54.53	55.44	69.62	41.78	63.76
TCASVM	78.45	79.31	63.35	70.05	71.00	50.48	45.23	68.72	36.88	62.61
JDASVM	80.54	81.02	68.94	73.60	93.13	66.95	29.25	46.17	54.02	65.59
DRCA	89.15	92.69	87.58	95.94	86.52	60.25	62.24	72.34	52.00	77.63
OSC	88.10	66.71	54.66	53.81	65.13	63.71	36.05	40.21	40.08	56.50
GFK	25.00	63.81	37.90	37.06	47.00	31.97	5.44	28.51	27.81	33.83
GLSW	78.38	69.36	80.75	74.62	69.43	44.28	48.64	67.87	46.58	64.43
DS	69.37	46.28	41.61	58.88	48.83	32.83	23.47	72.55	29.03	46.98
DANN+SAM	87.33	89.14	86.12	84.91	89.53	80.92	70.33	71.87	66.89	80.78
MADA	94.03	95.55	90.15	85.76	90.79	79.33	63.45	66.37	63.58	81.01
TWDDA	90.59	**97.79**	83.23	77.66	**97.87**	80.98	65.98	68.51	66.33	80.99
WDAAN	93.62	96.44	**90.37**	90.79	95.41	81.53	64.40	69.27	65.55	83.05
CDAN+SAM	**95.10**	96.21	88.81	**98.48**	95.86	**85.99**	**79.59**	**97.87**	**74.97**	**90.32**

**Table 3 sensors-24-01319-t003:** Recognition accuracy (%) under Dataset A.

Hyperparameters	1–2	1–3	1–4	1–5	1–6	1–7	1–8	1–9	1–10
SAM learning rate	0.001	0.001	0.001	0.001	0.001	0.001	0.001	0.001	0.001
Discriminator learning rate	0.01	0.01	0.01	0.01	0.01	0.01	0.01	0.01	0.01
Mini-batch size	8	16	16	32	32	64	32	32	32

**Table 4 sensors-24-01319-t004:** Data description of the complex E-nose data.

Batch	HCHO	C_6_H_6_	C_7_H_8_	CO	NO_2_	NH_3_	Total
Master	126	72	66	58	38	60	420
Slave1	108	108	106	98	107	81	608
Slave2	108	87	94	95	108	84	576

**Table 5 sensors-24-01319-t005:** Recognition accuracy (%) under Dataset B.

Method	Master-Slave 1	Master-Slave 2	Average
PCASVM	47.86	39.23	43.54
LDASVM	42.11	41.32	41.71
SVM-rbf	33.06	27.43	30.24
SVM-gfk	34.21	44.27	39.24
TCASVM	56.41	58.85	57.63
JDASVM	51.32	53.47	2.39
DRCA	57.07	52.95	55.01
DANN+SAM	55.23	56.74	55.99
MADA	50.04	55.72	52.88
TWDDA	59.38	56.08	57.73
WDAAN	61.39	57.70	59.54
CDAN+SAM	**67.60**	**71.52**	**69.56**

**Table 6 sensors-24-01319-t006:** Parameters’ values of the CDAN+SAM under Dataset B.

Hyperparameters	Master-Slave 1	Master-Slave 2
SAM learning rate	0.001	0.001
Discriminator learning rate	0.01	0.01
Mini-batch size	16	32

**Table 7 sensors-24-01319-t007:** Ablation study Accuracy (%) Under dataset A.

Method	1–2	1–3	1–4	1–5	1–6	1–7	1–8	1–9	1–10	Average
DANN+SAM	87.33	89.14	86.12	84.91	89.53	80.92	70.33	71.87	66.89	80.78
CDAN	90.95	90.54	79.68	92.89	94.48	93.02	72.63	94.25	70.41	86.53
CDAN+SAM	**95.10**	**96.21**	**88.81**	**98.48**	**95.86**	**85.99**	**79.59**	**97.87**	**74.97**	**90.32**

**Table 8 sensors-24-01319-t008:** Ablation study accuracy (%) under Dataset B.

Method	Master-Slave 1	Master-Slave 2	Average
DANN+SAM	55.23	56.74	55.99
CDAN	52.79	53.35	53.07
CDAN+SAM	**67.60**	**71.52**	**69.56**

## Data Availability

Data are contained within the article.

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
