# Peer review of "Electronic Nose Drift Suppression Based on Smooth Conditional Domain Adversarial Networks"

_sensors, 2024, doi:10.3390/s24041319_

Round 1

Reviewer 1 Report

Comments and Suggestions for Authors

H. Zhu et al. address the challenge of electronic nose drift by introducing CDAN+SAM, a deep transfer learning methodology. The approach utilizes feature extractors and domain discriminators to capture shared features from drift and clean data, enhancing model generalization. CDAN+SAM offers advantages such as semi-supervised learning, accommodation of inter-class correlations, and improved ease of training compared to conventional methods. Experimental results on two datasets demonstrate the methodology's efficiency and effectiveness in mitigating drift compared to state-of-the-art techniques. The paper is well organized and informative. The reviewer's comments are below:

1)  Could you provide more details on the reproducibility of the methodology? Specifically, are the hyperparameters, dataset details, and implementation specifics thoroughly documented to facilitate reproducibility by other researchers?

2) Considering the potential application of the proposed framework in real-world scenarios, could you discuss the scalability of CDAN+SAM? How does the computational complexity scale with larger datasets, and are there any practical considerations or constraints in terms of hardware requirements for implementing this methodology in a real-world sensor network?

3) Could you elaborate on the sensitivity of CDAN+SAM to different magnitudes of drift? Does the methodology exhibit consistent effectiveness across various levels of drift, and are there specific conditions where its performance might be more or less pronounced?

4) Does the methodology exhibit general applicability to different types of sensors, or is it tailored specifically to electronic nose data? 

5) Could you please provide insights into potential future directions for research building upon this work? 

According to the above comments, I recommend a major revision.

Good luck.

Comments on the Quality of English Language

N/A

Reviewer 2 Report

Comments and Suggestions for Authors

The paper presents interesting founding on algorithms for drift suppression. However some minjor revision needs to be done before publication.

First, too many abbreviations are used, making it very difficult for understanding. For example, CDAN and SAM are given in abstract without full name. Impossible to understand.  The same exists in the text.what does it mean of Ir in Table 3 and 6?

Second, no need to use abbreviations in Figures, such as P, F, C, D, FS,FT. Please use full name.

Thirdly, Figure 3 is impossible to read. Please adjust the size of notation.

Forthly, regarding figure 4 and 6, did you use 5-fold validation? Please add the error in the figure 4 and 6. The error and reliablity of the algorithms are needed.

Comments on the Quality of English Language

the english needs improvement. some errors like plural of noun and space (taken.carbon,formaldehyde in Line 2018, No spacing in keywords: Anti-drift,Electronic nose,Transfer learning,Deep learning) can be detected

Reviewer 3 Report

Comments and Suggestions for Authors

The introductory sentence of the abstract should give general information about the subject of the article. Unfortunately, it wasn't a good opening sentence. Edit the second sentence.

There must be a space after the period at the end of the sentence. Check out the full article.

Give the meanings of CDAN+SAM and DANN and MDANN in the summary.

The texts on the axes of the graphs in Figure 3 and Figure 5 are not readable.

In order to better understand this article, a flow diagram must be drawn for the processes performed.

I found the conclusions part of the study very inadequate.

At the end of the introduction section, the contributions of the study to the literature should be given in detail. “Chongqing University[40].” There must be a space before the reference. There are many errors like this in the article. Please check.

The article must be language checked.

Two datasets were used in the article. Accordingly, two different sets of results were obtained. These results are given comparatively. But it should be explained with reasons. So there should be more comments in the Results section. The study is not explained in detail. Frankly, what was said did not satisfy me.

Comments on the Quality of English Language

Language and spelling must be checked.

Round 2

Reviewer 3 Report

Comments and Suggestions for Authors

The requested changes have been made. The article is acceptable in its current state.